# An Integrative Analysis of the Immune Features of Inactivated SARS-CoV-2 Vaccine (CoronaVac)

**DOI:** 10.3390/vaccines10060878

**Published:** 2022-05-30

**Authors:** Zhujun Jiang, Haishuang Lin, Haoran Zhang, Ningning Shi, Zhetao Zheng, Liangzhen Dong, Yuelin Yang, Qing Xia

**Affiliations:** Department of Chemical Biology, State Key Laboratory of Natural and Biomimetic Drugs, School of Pharmaceutical Sciences, Peking University, Beijing 100191, China; jiangzhujun@bjmu.edu.cn (Z.J.); hlin9@pku.edu.cn (H.L.); henryzhang@bjmu.edu.cn (H.Z.); ning_0712@bjmu.edu.cn (N.S.); zhetaozheng@pku.edu.cn (Z.Z.); lzdong@bjmu.edu.cn (L.D.); yangyuelin@bjmu.edu.cn (Y.Y.)

**Keywords:** SARS-CoV-2, CoronaVac vaccine, immune features, serum-neutralizing antibody, multi-omics

## Abstract

Currently, an inactivated vaccine has been widely used with encouraging results as a prophylactic agent against COVID-19 infection, which is caused by severe acute respiratory syndrome coronavirus 2 (SARS-CoV-2) and its variants. However, in vitro SARS-CoV-2 vaccine-specific immune features remain elusive, hindering the promotion of a third dose of the vaccine. Here, we present a detailed in vitro immune cellular response and large-scale multi-omics analysis for peripheral blood mononuclear cells (PBMCs) from participants vaccinated with CoronaVac (Sinovac Life Sciences, Beijing, China) and recovered participants from COVID-19. The mean titers of SARS-CoV-2 serum-neutralizing antibodies were significantly increased after the boosting immunization (Day 45) compared to the unimmunized state. We observed that type-1 helper T cells (Th1) tended to dominate after the first dose of vaccine, while humoral immune responses became dominant after the second dose due to the activation of type-2 helper T cell (Th2), memory B cells, and plasmablasts. T follicular helper cells (Tfh) involved in antibody production were activated after the first dose and were maintained for the observed time points. Single-cell RNA sequencing of PBMCs revealed specific changes in cell compositions and gene expression in immunized participants. Multi-omics analysis also demonstrated that CoronaVac-specific serum proteins, plasma metabolites, and plasma lipid changes were skewed to those changes in convalescent patients. Collectively, we provide a comprehensive understanding of CoronaVac-specific in vitro immune features.

## 1. Introduction

Severe acute respiratory syndrome coronavirus 2 (SARS-CoV-2) has caused more than 268 million cases with over 5.2 million deaths worldwide as of 10 December 2021, according to the statistics of the World Health Organization. Due to the high infectivity and mutation rate of SARS-CoV-2 and a shortage of effective treatments, the rapid development of safe and efficacious vaccines is imperative [1]. To date, several SARS-CoV-2 vaccines have been approved by at least one country, including inactivated vaccines (e.g., Covaxin, QazVac, BBIBP-CorV, Inactivated (Vero Cells, Wuhan), ConoraVac, SARS-CoV-2 Vaccine (Vero Cells, Minhai Biotechnology Co.), COVID-19 Inactivated Vaccine (Shifa Pharmed Industrial Co.)), mRNA vaccines (e.g., mRNA-1273 and BNT162b2), protein subunit vaccines (e.g., CIGB-66, RBD-Dimer, and EpiVacCorona), and non-replicating viral vector (e.g., Ad5-nCoV, Sputnik V, Sputnik Light, Ad26.COV2.S, AZD1222, Covishield) [2] (https://covid19.trackvaccines.org/vaccines/, accessed on 10 December 2021). More than 8.2 billion vaccine doses have been administered worldwide as of 10 December 2021, of which over 2.5 billion people in mainland China have received the SARS-CoV-2 vaccine. As a widely used strategy against virus infection, the inactivated vaccines maintain the immunogenicity of the virus to induce protective immune responses. CoronaVac (initially known as PiCoVacc), an inactivated SARS-CoV-2 vaccine developed by Sinovac Life Sciences in China, has demonstrated good safety, immunogenicity, and efficacy in its clinical trials [3,4,5], and the two-dose vaccination is effective against the Delta variant infection, with an estimated efficacy exceeding the minimal threshold of 50% set by the World Health Organization [6]. Thus far, 26 clinical trials for CoronaVac have been performed in eight countries, and CoronaVac has been approved in 46 countries (https://covid19.trackvaccines.org/vaccines/, accessed on 10 December 2021). Although the neutralizing antibody titers decreased significantly in recipients within a few months of receiving the second dose of the vaccine, Xie et al. demonstrated that CoronaVac elicited a humoral immune response, including neutralizing antibodies, to SARS-CoV-2 variants, and suggested that a third-dose boost may be beneficial for combating SARS-CoV-2 and its variants [7]. However, the in vitro cellular immune features of CoronaVac are still unclear and require more thorough studies to demonstrate the important role the third-dose boost plays in further inducing effective immune memory.

Recently, large-scale multi-omics analyses [8,9,10], including single-cell RNA sequencing (scRNA-seq), proteomics, metabolomics, and lipidomics, have been used to illustrate the immune features between COVID-19 patients and healthy subjects [11,12,13,14]. These studies demonstrated that integrating multi-omics with an in vitro immune response analysis could be applied to evaluate the protective effects of a vaccine, thereby accelerating vaccine development [15]. In this study, we comprehensively evaluated the immune features of CoronaVac by stimulating PBMCs from vaccinated participants and convalescent participants in vitro and performing large-scale multi-omics analyses. The neutralizing antibody titers to CoronaVac or live SARS-CoV-2, the immune responses patterns, and the cytokine release activity of CD4+ T cells in PBMCs were monitored. We further investigated the single-cell landscape of PBMC samples from healthy participants at multiple timepoints before and after CoronaVac vaccinations and from control participants that had recovered from COVID-19 for approximately eight months. Proteomic, metabolomic, and lipidomic analyses were also performed for the serum or plasma samples in order to compare to the single-cell transcriptomics data. The aim of this study was to integrate in vitro immune assays and large-scale multi-omics analyses in order to reveal the immune characteristics of CoronaVac.

## 2. Materials and Methods

### 2.1. Study Design and Participants

In the vaccination group, 13 healthy adult or elder participants were enrolled. Key exclusion criteria include: high-risk epidemiology history within 14 days before enrollment; SARS-CoV-2 specific IgG or IgM positive in serum; positive PCR test for SARS-CoV-2 from a pharyngeal; axillary temperature higher than 37.0 °C; and known allergy to any vaccine component. In the recovery group, 12 adult or elder participants that had recovered from COVID-19 for approximately eight months were enrolled. All participants were recruited from Capital Medical University Youan Hospital, and their peripheral blood samples were collected with written informed consent. This study was approved by the Peking University Biomedical Ethics Committee and Youan Hospital (Approval number: LL-2020-041-K) and conducted in accordance with the requirements of Good Clinical Practice of China and the International Conference on Harmonization.

No specific randomization was used since only one vaccination group was involved. Comparisons were conducted between multiple timepoints of the vaccination group, or with the recovery control. Five participants withdrew at varied timepoints in the vaccination group for poor compliance or for getting bored with blood sampling (Figure 1), and their blood samples were no longer collected for the rest of the timepoints. Due to the high expense of single-cell RNA sequencing, only 11 samples from five randomly chosen adult participants (H1, H2, R1, R2 and R3) were used for scRNA-seq.

### 2.2. Inactivated SARS-CoV-2 Vaccine

CoronaVac was produced and assigned to clinical trials by Sinovac Life Sciences (Beijing, China). Briefly, African green monkey kidney cells (Vero cells) were inoculated with SARS-CoV-2 (CN02 strain). After replication, the virus was harvested, inactivated with β-propiolactone, concentrated, purified, and finally absorbed onto aluminum hydroxide. The aluminum hydroxide complex was diluted in a sodium chloride, phosphate-buffered saline, and water solution before being sterilized and filtered ready for injection. The vaccine was prepared in a good manufacturing practice-accredited facility of Sinovac that is periodically inspected by the Chinese National Medical Products Administration committee for compliance. Vaccines of 3 μg antigen in 0.5 mL of 225 μg aluminum hydroxide diluent per dose in ready-to-use syringes were administered intramuscularly.

### 2.3. PBMCs Collection

Healthy participants were injected with two doses of CoronaVac (3 μg per dose, at Day 0 and 28, right after blood sampling), with peripheral blood samples collected at Day 0, 14, 28, 35, and 45 to monitor dynamic changes. Peripheral blood samples were also collected from recovered participants as positive control. PBMC were isolated from heparinized peripheral blood samples (~20 mL) of healthy or recovered participants using a FicollR Paque Plus (Sigma Aldrich) solution with standard density gradient centrifugation methods. The cells were harvested and counted via a Cellaca MX high-throughput cell counter (Nexcelom Bioscience). The collected PBMC were either studied directly or cryopreserved in liquid nitrogen until used in the following assays.

### 2.4. Single-Cell RNA Sequencing

For scRNA-seq, frozen vials of PBMC from participants were rapidly thawed in 37 °C water bath for two minutes until a tiny ice crystal was left. Thawed PBMC were quenched with 4 mL 37 °C pre-warmed 1xPBS supplemented with 10% fetal bovine serum. Cells were centrifuged at 500 g for 10 min at room temperature and the supernatant was discarded. The cell pellet was resuspended in 3 mL 1xPBS containing 0.04% bovine serum albumin and centrifuged. Dead cells were removed by magnetic bead purification (Miltenyi Biotech). Finally, cells were resuspended at a viable cell concentration of 1000 cells/μL. After droplet encapsulation, emulsion breakage, mRNA captured bead collection, reverse transcription, cDNA amplification, and purification, the single-cell suspensions were converted to barcoded scRNA-seq libraries. The sequencing libraries were quantified by Qubit ssDNA Assay Kit (Thermo Fisher Scientific) and sequenced by a NovaSeq 6000 sequencer at Beijing Emei Tongde Technology Development Co., Ltd., Beijing, China. The single-cell RNA sequencing data were analyzed by Cell Ranger (version 5, 10X Genomics, Shanghai, China) and Loupe Browser (version 5, 10X Genomics, Shanghai, China).

### 2.5. Neutralizing Antibodies Detection

After collecting all samples, the serum-neutralizing antibodies to ConoraVac at indicated timepoints before and after vaccination and live SARS-CoV-2 (virus strain SARS-CoV-2/human/CHN/CN1/2020, GenBank number MT407649.1) were quantified using a micro cytopathogenic effect assay with a minimum four-fold dilution. The antibody detection tests were performed by Sinovac Life Sciences (Beijing, China).

### 2.6. Immunophenotypic Analysis of PBMCs

LIVE/DEAD™ Fixable Aqua Dead Cell Stain (L34957, Thermo Fisher) was used to exclude dead cells. CD1c (Biolegend), CD141 (Biolegend) were used for DCs analysis by flow cytometry. Fluorochrome-labeled anti-human mAbs specific for CD3 (UCHT1, Biolegend), CD4 (OKT4, Biolegend), CD45RA (HI100, Biolegend), CXCR5 (J252D4, Biolegend), CCR6 (G034E3, Biolegend), CXCR3 (G025H7, Biolegend), PD-1 (EH12.2H7, Biolegend) and ICOS (C398.4A, Biolegend) were used for analysis of Tfh cell population by flow cytometry. Fluorescein-labeled anti-human antibodies specific for CD3 (UCHT1, Biolegend), CD19 (HIB19, Biolegend), CD21 (Bu32, Biolegend), CD27 (O323, Biolegend), CD38 (HIT2, Biolegend) and IgD (IA6-2, Biolegend) were used for analysis of B cell population by flow cytometry.

### 2.7. Cytokines Release Analysis

For the cytokine release activity assay, 1 × 10^6^ CD4+ T cells were purified from PBMC by positive selection using magnetic-activated cell sorting (MACS, Miltenyi Biotec) and then stimulated with or without 6 μg/mL CoronaVac (no aluminum hydroxide) in RPMI 1640 medium (Corning) supplemented with 10% fetal bovine serum (FBS, VWR), 1% penicillin (Gibco), and 2 μg/mL anti-CD28 antibody for 72 h. The supernatants were collected, and 12 human cytokines (IFN-γ, TNF-α, IL-2, IL-4, IL-5, IL-6, IL-9, IL-10, IL-13, IL-17A, IL-17F, and IL-22) were detected by two sets of capture beads and quantified by Beckman Coulter CytoFLEX LX using the LEGENDplex™ Multi-Analyte Flow Assay kit (741028, Biolegend). The data were analyzed by LEGENDplex™ Data Analysis Software. For 67 samples from 25 participants, the serum proteomics (59/67), plasma metabolomics (64/67), and plasma lipidomics (67/67) were also detected by mass spectrometry, although some samples were exhausted and absent in some experiments.

### 2.8. Multi-Omics Analyses of Blood Samples

For the vaccination group, 55 peripheral blood samples were collected in total, covering 13 participants and five timepoints. For the recovery group, 12 peripheral blood samples were collected. PBMC from 11 samples (eight from the vaccination group and three from the recovery group) were subjected to single-cell RNA sequencing, in which around 10,000 cells were analyzed and 50–90 gigabytes data were generated for each PBMC sample. The 11 scRNA-seq results were aggregated into one cloupe file, and the main cell types were identified by gene markers (B cells by MS4A1; CD4+ T cells by CD3G and CD4; CD8+ T cells by CD3G and CD8B; NK cells by KLRF1; Myeloid cells by LYZ) with reference to graph-based clusters. The t-SNE clustering, cell number and percent, and differential gene expressions were analyzed to illustrate the influences of vaccination on PBMC transcriptomics. After collecting all 67 samples, their serum-neutralizing antibody titers to live SARS-CoV-2 were measured. For the cytokine release activity, 63 samples were actually assayed, since four samples (H12_Day 0, H11_Day 14, H12_Day 14, and H3_Day 14) did not collected enough cells for all assays. The serum proteomics covered samples of Day 0, 14, 28, 35, and Month 8, while plasma metabolomics and lipidomics covered samples of all timepoints.

### 2.9. Statistical Analysis

Quantitative data were visualized and statistically analyzed in GraphPad Prism (version 8). For the multiple-timepoint immune cell percent in total PBMC of the two participants in the vaccination group, one-way RM ANOVA and Tukey’s multiple comparisons test were performed. For the serum-neutralizing antibody titers of all samples, Kruskal–Wallis test and Dunn’s multiple comparisons test were performed. For the cytokine release activity assays of all samples, multiple t tests were performed between stimulated and unstimulated groups. A *p* value smaller than 0.05 was considered statistically significant. For summary of the *p* value, # *p* < 0.1; * *p* < 0.05; ** *p* < 0.01; *** *p* < 0.001; **** *p* < 0.0001. For expression data of proteomics, metabolomics, and lipidomics, principal component analysis (PCA) was performed.

### 2.10. Data Availability

Single-cell RNA sequencing data and report were uploaded to the figshare repository (DOI: 10.6084/m9.figshare.19808818). All other data are available from the corresponding authors on request.

## 3. Results

### 3.1. Study Design and Serum-Neutralizing Antibody Titer Assay

Twenty-five participants were enrolled in the study, including 13 healthy participants vaccinated with two doses of CoronaVac and 12 participants that recovered from COVID-19 (Figure 1a). All samples from 25 participants were assayed for serum-neutralizing antibody titers, cell immune patterns, and cytokine release activity of CD4+ T cells. Most of the samples (>80%) from 25 participants were subjected to serum proteomic, plasma metabolomic, and lipidomic analyses. Representative participants H1 and H2 (each with four timepoints) from eleven PBMC samples and R1, R2, and R3 (each with one timepoint) were subjected to single-cell RNA sequencing. We measured the serum-neutralizing antibody titers to CoronaVac and live SARS-CoV-2 for all 25 participants (Figure 1b). Similar to previous clinical data (NCT04352608), the neutralizing antibody titers showed a time-dependent increase and became significant at Day 35 (19.8) and Day 45 (24.2) (i.e., 7 and 17 days after the second vaccination), although they were still lower than the recovery control (126.9) that once suffered systemic SARS-CoV-2 infection. The second vaccination caused stronger effects than the first, which could be attributed to memory B cells. The mean titers of SARS-CoV-2 serum-neutralizing antibodies on day 45 were 11-fold in immunized participants and 57.7-fold in recovered participants compared to unimmunized participants.

### 3.2. The Specific Immune Response Patterns of CoronaVac

In order to evaluate the specific immune responses of two doses of CoronaVac at indicated timepoints, we analyzed dendritic cells (DCs), T follicular helper (Tfh) cells and their subtypes, and B cells and their subtypes in PBMCs by flow cytometry (Figure 2a,b). We first quantified the DC response, which plays a central role in the initiation and shaping of the primary immune response [16]. An increase in CD1c+ DCs in the PBMCs was observed on Day 14 after the first vaccination, but not CD141+ DCs (Figure 2c). Circulating CXCR5+CD4+ T Cells are counterparts of T Follicular cells. They can be classified into multiple subsets depending on the expression of CXCR3, CCR6, PD-1 and ICOS1 [17,18]. We then quantified circulating Tfh cells and their subsets, which have distinct phenotype and functions in supporting anti-viral antibody secretion [19]. We observed a significant expansion of circulating Tfh cells (CXCR5+CD45RA−) in PBMCs at Day 14 after the first vaccination, which maintained to the end of the observation period (Day 45). Th1-like Tfh cells (CXCR3+CCR6−), which are essential for the maintenance of long-term neutralizing antibody responses [20], and circulating PD-1+/CXCR3-/CXCR5+ memory Tfh cells, which correlate with the generation of broadly neutralizing antibodies [21], were significantly increased on Day 14 after the first vaccination and were maintained to Day 35. Th2-like Tfh cells (CXCR3−CCR6−), which are associated with the functional breadth and magnitude of parasite antibodies [22], were only activated during the early stage of vaccination. Th17-like Tfh cells (CXCR3−CCR6+), which induce naïve B cells to produce immunoglobulins and undergo isotype switching [23], were significantly activated after the second vaccination and maintained to Day 35. CXCR3+ICOS+ Tfh cells, which induce memory B cells [24], were significantly increased on Day 35 and were maintained to a high level on Day 45 (Figure 2d). For B cell responses, an increase in circulating memory B cells and plasmablasts was detected after vaccination, indicating activation of these cells, which are responsible for the generation of antibodies (Figure 2e). However, we did not observe significant changes in CD8+ T cells, which might be due to the low level of these cells at the tested time points or more sensitive methods are needed to detect these cells. Collectively, we demonstrated that the CoronaVac could elicit a strong immune response, including immune priming and antibody generation.

### 3.3. Cytokine Release Activity of CD4+ T Cells

CD4+ T cells play a crucial role in protection against pathogens and eliciting immune responses [25]. To obtain phenotypic features of CoronaVac-induced CD4+ T cells, CD4+ T cells were purified from PBMCs via magnetic-activated cell sorting and stimulated them with or without 6 μg/mL CoronaVac (excluding its Al(OH)3 adjuvant, Figure 3a). The levels of 12 cytokines released into the supernatant were detected (Figure 3b–m). Five cytokines showed significantly increased release after stimulation, including interferon-γ (IFN-γ), interleukin-2 (IL-2), interleukin-4 (IL-4), interleukin-6 (IL-6), and interleukin-10 (IL-10). As type-1 helper T cell (Th1) cytokines [26], IFN-γ and IL-2 were mainly activated at Days 14, 28, and 35, supporting cellular immune responses during the early stages of vaccination (Figure 3b,c). As type-2 helper T cell (Th2) cytokines [27], IL-4, IL-6, and IL-10 were mainly activated at Days 28, 35, and 45, supporting humoral immune responses during the later stages, especially after the second vaccination (Figure 3e–g). Other cytokines, such as TNF-α, IL-5, IL-9, IL-13, IL-17A, IL-17F, and IL-22, did not generate significant differences after stimulation, indicating that CoronaVac induces minimal adverse events associated with inflammation (Figure 3d,h–m). In general, mixed Th1/Th2 responses were observed in the CD4+ T cells from participants after CoronaVac vaccination. Although small samples were used due to the cost limit, we still observed some significant differences, which could help identifying the immune response patterns of CoronaVac. These data addressed the in vitro immune response patterns and could complement the clinical trials of CoronaVac performed in China [3,28,29] (NCT04383574), Brazil [3,30] (NCT04456595), Indonesia [3] (NCT04508075), and Turkey [31] (NCT04582344).

### 3.4. scRNA-seq of PBMCs

In order to systematically investigate the immune characteristics at the single-cell level, we performed scRNA-seq on PBMCs collected from unvaccinated, vaccinated, and convalescent participants (Figure 4a). Due to the high cost of scRNA-seq, we cut down the sample size to two vaccination participants (H1 and H2, each with four time points) and three recovery participants (R1, R2 and R3). PBMCs were isolated from the 11 samples, and around 10,000 cells were analyzed by scRNA-seq for each PBMC sample (Figure 4b), constituting a total of 103,931 cells. The t-distributed stochastic neighbor-embedding (t-SNE) plots were used to identify cell clusters and sample similarities [31]. By categorizing and coloring the points according to the sample information (Appendix A), the similarity of the cell compositions and gene expressions of each sample could be evaluated. Two PBMC samples (H1_Day 0 and H2_Day 0) from the unvaccinated healthy participants overlapped vastly on the t-SNE graph, indicating that healthy participants possessed similar PBMC transcriptomics. The samples could be further aggregated by the observation time, such as combining H1_Day 0 and H2_Day 0 into Day 0, which enabled comparisons through the vaccination timeline. It was also convenient to combine Days 14, 28, and 35 into one stage (i.e., after vaccination). We found that samples before vaccination showed distinct clustering from those after vaccination or the recovery control (Appendix A), while samples of the latter two groups demonstrated similar clustering (Appendix A). Despite the small sample size, scRNA-seq was a sensitive method for conveying multidimensional gene expression profiles and could reveal the subtle transcriptomic changes before and after vaccination according to the t-SNE clustering (Appendix A). We concluded that vaccinations with CoronaVac in healthy participants altered the PBMC transcriptomics into a state resembling the COVID-19 recovery stages, which seemed to be positive immune responses.

To detail the immune responses in PBMC transcriptomics, we classified the constituent cells of PBMCs using specific gene markers [12] expressed by B cells, CD4+ T cells, CD8+ T cells, NK cells, and myeloid cells (Figure 4c,d). These cell types formed four main clusters on the t-SNE graph, among which CD4+ and CD8+ T cells partially overlapped (Figure 4e). The percent of each cell type in total PBMCs was analyzed through the vaccination timeline with recovery as control (Figure 4f). We found that the percent of B cells increased from Day 0 to 14 and from Day 28 to 35 after each vaccination, but decreased from Day 14 to 28, implying that the B cell level relied on vaccinations or external antigen stimulation. Moreover, the B cell level could reach that of the recovery control after each vaccination, which was further proof of a positive immune response. The percent of T cells exhibited individual variance [32] between the two participants (H1 aged 21 years and H2 aged 42 years), but the trends within an individual were consistent. Both CD4+ and CD8+ T cells increased from Day 0 to 28, peaked at Day 28, and then decreased from Day 28 to 35. Participant H1, which had lower levels of CD4+ T cells, tended to have higher levels of CD8+ T cells than participant H2, suggesting a balance between humoral and cellular immunity. Natural killer (NK) cells showed an inverse trend to B cells, while myeloid cells showed an inverse and drastically different trend to T cells. The percent changes of different cell types confirmed that vaccinations with CoronaVac altered the cellular composition of PBMCs.

According to the detailed t-SNE graph (Figure 4e), Day 0 clustered separately from other timepoints and the recovery control for each cell type, indicating that the gene expressions in B cells, T cells, NK cells, and myeloid cells changed after CoronaVac vaccination or recovery from COVID-19. In addition, gradual transitions were observed from Day 14 to 28 and 35 in the vaccination groups, which finally approached Month 8 of the recovery control. The expression heatmap of significant genes for 25 subclusters (Figure 4g) supported that vaccinations with CoronaVac altered the gene expression of PBMCs. With appropriate gene markers, it was possible to classify PBMCs into more subtypes (Appendix A). The unvaccinated samples (Day 0) clustered separately from the vaccinated or recovered samples, while the vaccinated samples gradually approached the recovered samples along the vaccination timeline (Appendix A), demonstrating that CoronaVac induced positive gene expression responses in PBMCs.

During the immune process, B cells produce antibodies for invading antigens, and we analyzed the differentially expressed (DE) genes in B cells after CoronaVac vaccination or SARS-CoV-2 infection (Appendix A). Compared with the B cells of unvaccinated participants, 33 DE genes were discovered after vaccination, while 62 DE genes were discovered in the recovery control, which implied that CoronaVac could keep part of the SARS-CoV-2 antigenicity and elicit partial responses judged by the DE gene number. These DE genes typically included some transcription factors and immunoglobulin fragments, which were considered to be related to CoronaVac vaccination or SARS-CoV-2 infection. It is interesting that 23 common DE genes emerged in both the vaccinated participants (23/33) and the recovery control (23/62), including 15 common upregulated genes and eight common downregulated genes, which may be essential to the generation of the SARS-CoV-2 antibody. Among the 15 common upregulated genes, eight genes were associated according to the STRING database and formed the JUN/FOS interaction network composed of transcription factors (e.g., AP-1) that can regulate cytokines and modulate the sequential recruitment and activation of immune cells [33,34]. We inferred that the JUN/FOS network played an essential role in the response of B cells to SARS-CoV-2 and might be a potential indicator for effective SARS-CoV-2 vaccines. Other common changes included tuned expressions of immunoglobulin fragments, such as IGHV3-30 and IGLV2-23, which may be the core fragments against the SARS-CoV-2 antigen. In addition, B cells of vaccinated participants only showed 13 DE genes as compared with the recovery control (Appendix A), which was much smaller than the former two comparisons, indicating that vaccination with CoronaVac altered the gene expression profiles of B cells toward that of the COVID-19 recovery control. More information about the DE genes, enrichment Gene Ontology (GO) terms, and KEGG analyses between vaccination groups in B cells, T cells (total, CD4+ or CD8+), NK cells and myeloid cells can be found in Appendix A.

### 3.5. Multi-Omics Analysis for the Immune Characteristics of CoronaVac

Recently, multi-omics analysis has been used to uncover functional connections or co-regulation between different classes of biomolecules [8,10]. To understand the immune characteristics of CoronaVac, we performed multi-omics analysis, including human serum proteomics (Figure 5), plasma lipidomics (Figure 6a,b), and plasma metabolomics (Figure 6c,d), which indicate immune changes after virus infection or vaccination. Proteomics analysis showed that 132 serum proteins had significant changes, among which 42 were upregulated before the vaccine and 90 were upregulated after the vaccine (Figure 5a–d), especially in the recovery patients, including C1QC, C1RL, INS-IGF2, CFHR1, TTR, and immunoglobulin heavy chain proteins (IGHV3-74, IGHV4-4, IGHV3-49, IGHA2) (Figure 5a,e). Serum protein levels during the late stage of vaccination (on Day 35 and later) were similar to those of the convalescent participants, indicating that CoronaVac could induce similar serum protein changes (Figure 5a,b). The hierarchical genome-wide overview of the reactome of serum proteins in vaccinated participants was mainly enriched in the immune system (Figure 6a). The 25 most relevant pathways sorted by *p* value were significantly enriched in complement-related pathways (Figure 6b), which play a crucial role in eliminating invading pathogens during the early stage of infection and are major contributors to the acute phase response [10]. Of the complement-related pathways, we found that 15 serum proteins had significant changes, among which two (IGLV1-36, CFHR4) were downregulated and 13 (IGLC7, IGLV2-11, CFHR1, C1QA, IGKV3-11, MASP2, IGKV3D-20, MASP1, IGKV4-1, C3, PROS1, C8B, C1S) were upregulated after the vaccine (Figure 6c). In addition, we found that calprotectin (S100A8 and S100A9) was mildly upregulated in the TLR pathway induced by endogenous ligands (Figure 6d), which was also reported in previous studies [9,10]. We also found several proteins (e.g., KLKB1, F12) that were negatively correlated with angioedema (Figure 6e).

The lipidomics analysis revealed that there was a significant decrease in phosphatidylglycerol (PG) and lysophosphatidylcholine (LPC) (Figure 7a,b), a known inhibitor of the viral membrane fusion, which can prevent the entry of the virus into the host cells, indicating similar LPC change with virus infection. LPC can also induce the migration of lymphocytes and macrophages, increase the production of pro-inflammatory cytokines, and aggregate inflammation, which could increase the body’s inflammatory response level to resist virus invasion [35]. However, a significant increase in phosphatidylcholine (PC), phosphatidylethanolamine (PE), phosphatidylserine (PS), and monogalactosyldiacylglycerol (MGDG) was observed after vaccination (Figure 7b). Those above changes were consistent with recovered patients, especially both at the late stage of vaccination and the recovered group. The results showed that the CoronaVac could induce similar lipids changes as the recovery participants. In the metabolomics analysis, both LPC and PC changes displayed consistent lipidomic changes. In addition, bile acids, cholestane steroids, carbohydrates, tricarboxylic acid cycle-related metabolites, and ketoacids-related metabolites showed a decreasing trend after vaccination and in convalescent participants, while amines, purine metabolites, and bilirubin increased in the late stage of vaccination and the recovery group (Figure 7c,d). The results showed that CoronaVac could induce similar metabolomics changes as compared to the convalescent group. Taken together, we utilized multi-omics approaches to not only uncover functional connections between the vaccine and the immune system, but also to demonstrate that CoronaVac-associated multi-omics changes could further reveal comprehensive immune characteristics of CoronaVac.

## 4. Discussion

In this study, we aimed to focus on the early immune response and mechanism of SARS-CoV-2 vaccine (CoronaVac). We depicted a multi-timepoint immune response and large-scale multi-omics atlas for the inactivated SARS-CoV-2 vaccine, CoronaVac, which is useful for illustrating the mechanisms and guiding the schedule of SARS-CoV-2 vaccines. The early immune response (within a month after the first vaccination) tended to include cellular immune responses, as demonstrated by low serum-neutralizing antibody titers (Figure 1b), high Th1 cytokine release activity (Figure 3), and increased CD8+ T cell levels (Figure 4). Judging from the CD4+ T cell levels that peaked at Day 28 and the fact that helper T cells may aid the humoral immune response, we recommend performing the second vaccination between Days 14 and 28, with Day 28 preferred, but no later than Day 28. After the second vaccination, Th2 cytokines, such as IL-4, IL-6, and IL-10, were significantly released, CD8+ cytotoxic T cell levels decreased, B cell levels increased, and the serum-neutralizing antibody titers also increased, demonstrating a tendency for humoral immune responses. A balance seemed to exist between cellular and humoral immune responses, which dominated at early and late stages of vaccination, respectively. The high serum-neutralizing antibody titers and distinct PBMC transcriptomics of the Month 8 recovery control contrasted to unvaccinated healthy participants, implying that human immunity to SARS-CoV-2 could sustain at least eight months, which supported long-term resistance to SARS-CoV-2, which is consistent with previous studies [36]. While scRNA-seq demonstrated increased B cell levels and altered PBMC transcriptomics from Day 0 to 14 after the first vaccination, the serum-neutralizing antibody titers from Day 14 and 28 were not significantly higher than those before vaccination, implying that scRNA-seq was more sensitive than antibody titer assays. By scRNA-seq of PBMCs and immune function assays, we found that vaccination with CoronaVac could induce positive immune responses in terms of PBMC transcriptomics, serum-neutralizing antibody titers, and Th1/Th2 cytokine release features.

Additionally, we demonstrated that large-scale multi-omics analysis could be used as an informative tool to monitor the immune characteristics of SARS-CoV-2 vaccines due to being able to comprehensively study changes in the protein, lipids, and metabolites, which complements scRNA-seq. Although several previous studies have demonstrated the detailed functional connections or co-regulations between different classes of biomolecules (e.g., protein, lipids, and metabolites) in COVID-19 patients or COVID-19 convalescent patients [8,9,11,13,14,37], there is no other study reporting these changes in vaccinated participants as compared to the COVID-19 convalescent control. Our study uncovered for the first time the comprehensive changes in genes, protein, lipids, and metabolites in vaccinated participants as compared to healthy controls and COVID-19 convalescent controls (Figure 4, Figure 5 and Figure 6). Vaccination with CoronaVac was able to transform the cellular omics into a state similar to the COVID-19 recovery control as compared to the omics state prior to vaccination, which indicated that the vaccine was working. Moreover, multi-omics analyses provide insights into the essential genes, protein, lipids, and metabolites changes caused by SARS-CoV-2 in any immune cell subtype. Judging from these results, we recommend multi-omics analyses of PBMCs compared to recovery controls as a routine evaluation for new vaccines. By integrating and analyzing the multi-omics data from healthy participants vaccinated with different SARS-CoV-2 vaccines, or recovered participants or COVID-19 patients of different SARS-CoV-2 mutants, it is also possible to compare vaccine performance and forecast the cross-protection of vaccines against SARS-CoV-2 mutants.

The neutralizing antibody titers after the second vaccine dose decreased significantly within a few months, and the recent rapid spread of the Delta variant worldwide, which has partially nullified vaccination [38] against SARS-CoV-2, forcing many countries to consider application of a third dose [39,40]. Xie et al. demonstrated that humoral immune response of the plasma and neutralizing antibodies elicited by CoronaVac to SARS-CoV-2 variants, such as B.1.351, also known as 501Y.V2, suggest that a third-dose boost may be beneficial for combating SARS-CoV-2 and its variants when necessary [7]. They used an extended gap (more than 6 months) between the second and third doses, which resulted in better neutralizing activity and tolerance to 501Y.V2 than the standard three-dose administration with at least a nine-month gap. Although the mutant strains nullified some of the neutralizing antibodies, Xie et al. developed a full spectrum of single neutralizing antibody drugs (DXP-604) that are resistant to all existing and new variants that might emerge in the future. In addition, although CoronaVac was not effective against Beta and Omicron variant of SARS-CoV-2 (https://www.rojakpot.com/pfizer-vs-sinovac-omicron-variant/, accessed on 22 May 2022), the vaccine against Omicron variant by Sinovac Life Sciences has been developed and approved on 26 April 2022.

There are several limitations in this study. First, the vaccinated time was focused on early-stage immune responses and gene expression changes of vaccinated healthy participants (45 days), which was insufficient to observe long-term neutralizing antibodies and immune features, while the control group was participants recovered from COVID-19 for 8 months. Second, the sample size, including recruited participants and scRNA-seq, was comparatively small. Third, detailed clinical data of COVID-19 patients, including severity of infection, as well as age and gender, should be analyzed. Future work is important for fully understanding the diverse biological and immunological patterns underlying these heterogenous patient populations. Lastly, we do not know if ConoraVac is protective against multiple SARS-CoV-2 variants, even if the two-dose vaccination of CoronaVac is effective against Delta variant infection with the estimated efficacy exceeding the World Health Organization minimal threshold of 50%. Given these limitations, further studies need to be performed in the future.

## Figures and Tables

**Figure 1 vaccines-10-00878-f001:**
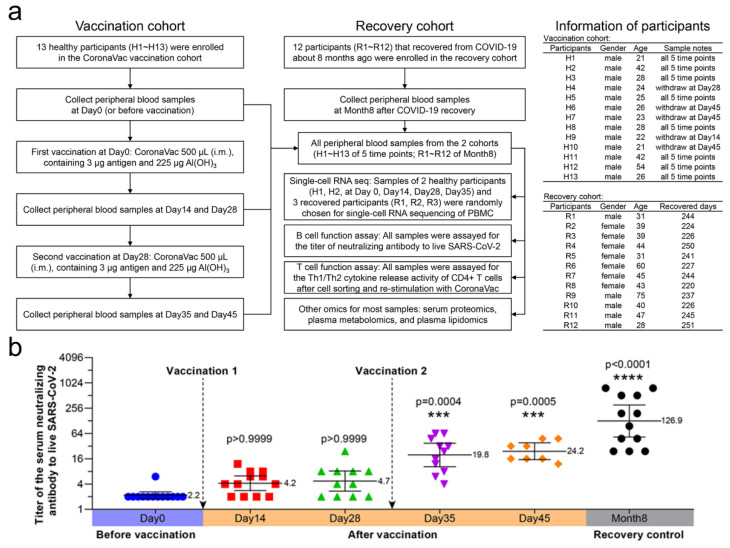
Study profile and the neutralizing antibody titers of vaccinated and recovery participants. (**a**) Diagram showing the number of, and information regarding, participants in the study. Samples from 25 participants were assayed for serum-neutralizing antibody titers to live SARS-CoV-2 and cytokine release activity of CD4+ T cells. The participant information is shown on the right, including gender, age, withdrawing stage and recovered days. The samples from 25 participants were subjected to serum proteomics, plasma metabolomics, and lipidomics. After balancing the cost of scRNA-seq and the sample size of each group, participants H1 and H2 (each with four timepoints) and R1, R2, and R3 (each with one timepoint) were randomly chosen from each cohort, which made up 11 samples for scRNA-seq. (**b**) Titers of serum-neutralizing antibodies to live SARS-CoV-2 (*n* = 8–13 in each group). The solid lines represent a geometric mean with 95% confidence interval. Kruskal–Wallis test and Dunn’s multiple comparisons test were performed. Titers were compared with that of Day 0 or before vaccination. The geometric mean, *p* value, and *p* summary are annotated on the graph. *** *p* < 0.001; **** *p* < 0.0001.

**Figure 2 vaccines-10-00878-f002:**
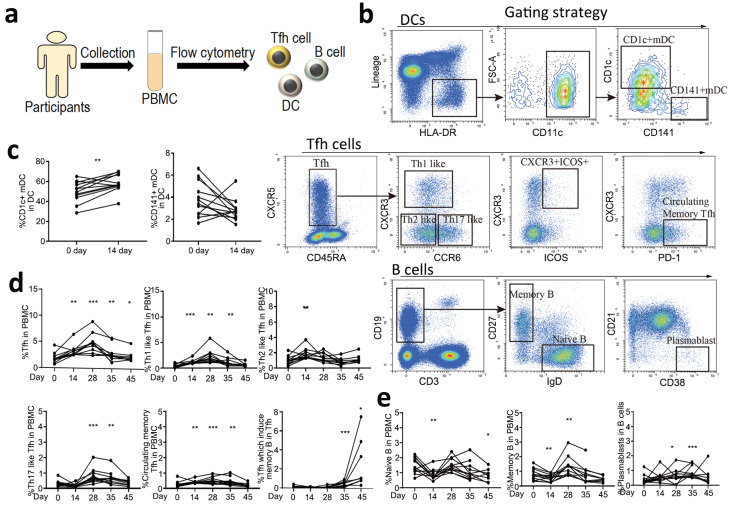
Specific immune response patterns of SARS-CoV-2 vaccine. (**a**) Flowchart of the cellular immune patterns assay. PBMCs were collected from 13 healthy participants, and DCs, Tfh cells, and B cells were analyzed by flow cytometry. (**b**) Gating strategies of flow cytometry. (**c**) The response of DCs before and after the first vaccination at Day 0 and 14. Day 0 (before vaccination) was as the control. Two tailed paired t test was used. (**d**) The response of Tfh cells, including Tfh cells, Th1-like Tfh cells, Th2-like Tfh cells, Th17-like Tfh cells, circulating memory Tfh cells, and Tfh cells, which induce memory B cells before and after ConoraVac vaccination at Day 0, 14, 28, 35, and 45. (**e**) The response of B cells, including naïve B cells, memory B cells, and plasmablasts before and after ConoraVac vaccination at Day 0, 14, 28, 35, and 45. For panels d and e, one-way RM ANOVA and Tukey’s multiple comparisons test were performed. Significance were annotated with Day 0 (before vaccination) as the control. ** p* < 0.05, ** *p* < 0.01, *** *p* < 0.001.

**Figure 3 vaccines-10-00878-f003:**
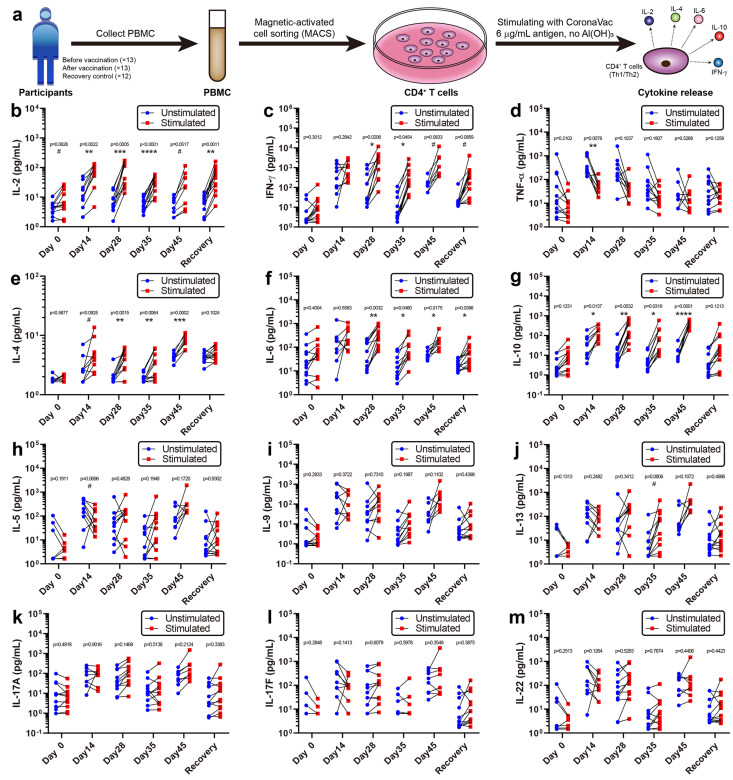
Cytokine release activity of CD4+ T cells. (**a**) Flowchart of the cytokine release activity assay. PBMCs were collected from all 25 participants, CD4+ T cells were isolated by MACS, and then one million CD4+ T cells per well were stimulated with CoronaVac, containing 6 μg/mL antigen but no Al(OH)_3_ adjuvant. Finally, the various cytokines released by CD4+ T cells were examined, and the mixed Th1/Th2 responses were observed. (**b**–**m**) Released cytokine concentrations were plotted (*n* = 8–13). IL-2 and IFN-γ are Th1 cytokines that functioned during the early stages (Days 14, 28, and 35), while IL-4, IL-6, and IL-10 are Th2 cytokines that functioned at the later stages (Days 28, 35, and 45). Other cytokines did not generate a significant response. Multiple *t* tests were performed between stimulated and unstimulated groups with the *p* value and p summary reported. # *p* < 0.1; * *p* < 0.05; ** *p* < 0.01; *** *p* < 0.001; **** *p* < 0.0001.

**Figure 4 vaccines-10-00878-f004:**
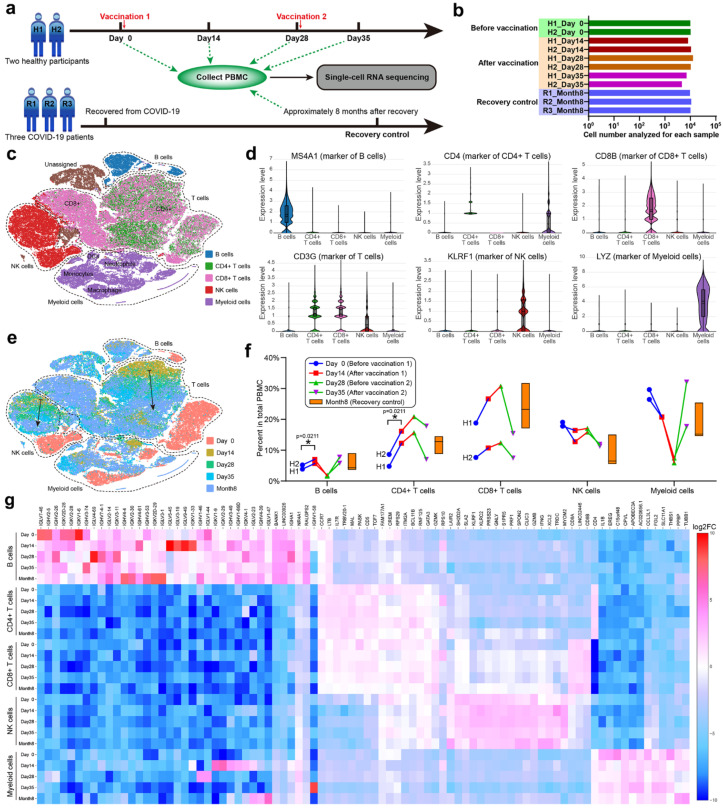
scRNA-seq of PBMC. (**a**) Group design and timeline for scRNA-seq of PBMCs. Eleven samples from five randomly chosen adult participants (H1, H2, R1, R2 and R3) were used for scRNA-seq. Two healthy participants were vaccinated with two doses of CoronaVac (3 μg) after collecting blood samples at Day 0 and 28. Three COVID-19 recovered participants were enrolled. PBMCs were isolated from 11 blood samples and sent for scRNA-seq. (**b**) The total cell number analyzed for each sample. (**c**) The main immune cell types in the isolated PBMCs. B cells, CD4+ T cells, CD8+ T cells, NK cells, and myeloid cells were identified by their gene markers and colored on the t-SNE plot. (**d**) Violin plots of representative gene marker expressions. (**e**) The t-SNE plot colored by the observation time. The dashed lines indicate clusters of four main cell types, and the two solid arrows indicate gradual transitions from Day 14 to 28, Day 35 of vaccination groups, and Month 8 of the recovery controls. (**f**) Percent of each cell type in the total PBMCs. The floating bars represent min, median, and max for the three participants in the recovery group. The symbols with connecting lines indicate multiple timepoints for the two participants in the vaccination group. Two-way RM ANOVA and Tukey’s multiple comparisons test were performed for the vaccination group. * *p* < 0.05. (**g**) A heatmap showing the expression of significant genes for the 25 subclusters generated by crossing cell types and observation time.

**Figure 5 vaccines-10-00878-f005:**
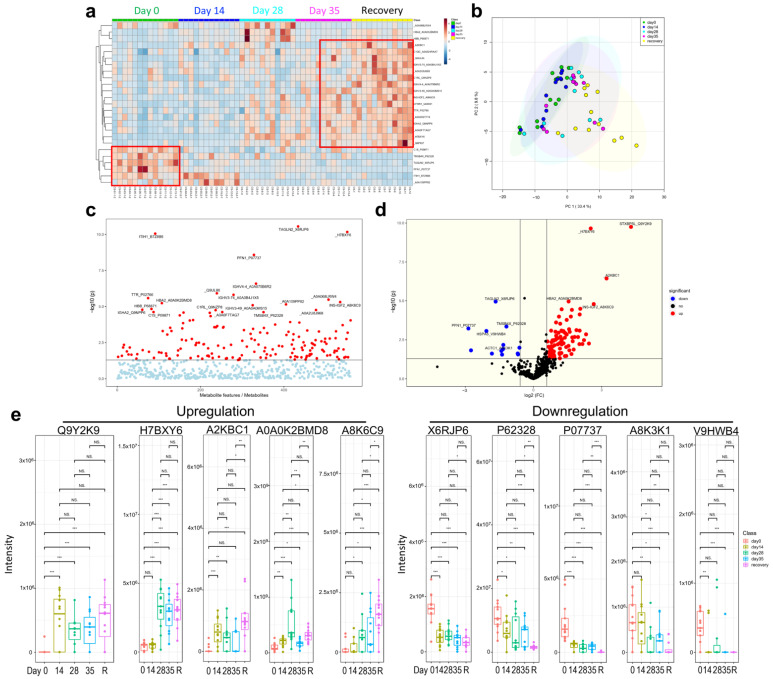
Human serum proteomics of vaccination-related or COVID-19-recovered samples. (**a**) Heatmap. (**b**) Principle component analysis (PCA). (**c**) Analysis of variance (ANOVA) for proteins. (**d**) Volcano plot. (**e**) Downregulated and upregulated proteins. NS: not significant; * *p* < 0.05; ** *p* < 0.01; *** *p* < 0.001.

**Figure 6 vaccines-10-00878-f006:**
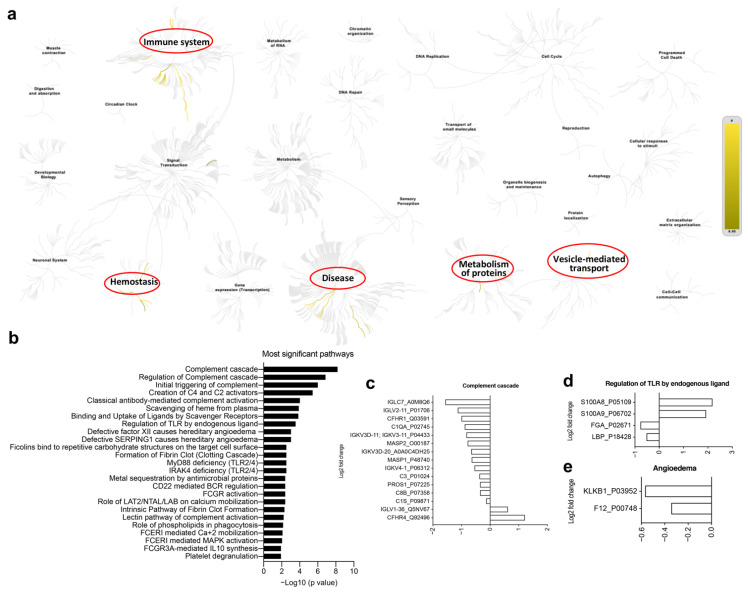
Human serum proteomics of vaccination-related or COVID-19-recovered samples. (**a**) The hierarchical genome-wide overview of the reactome of serum proteins in vaccinated participants, which were significantly enriched in immune system, hemostasis, disease, metabolism of proteins and vesicle-mediated transport. (**b**) The 25 most relevant pathways, sorted by *p* values, were significantly enriched in complement-related pathways. (**c**–**e**) Differential gene expression of complement cascade (**c**), regulation of TLR by endogenous ligands (**d**) and angioedema (**e**).

**Figure 7 vaccines-10-00878-f007:**
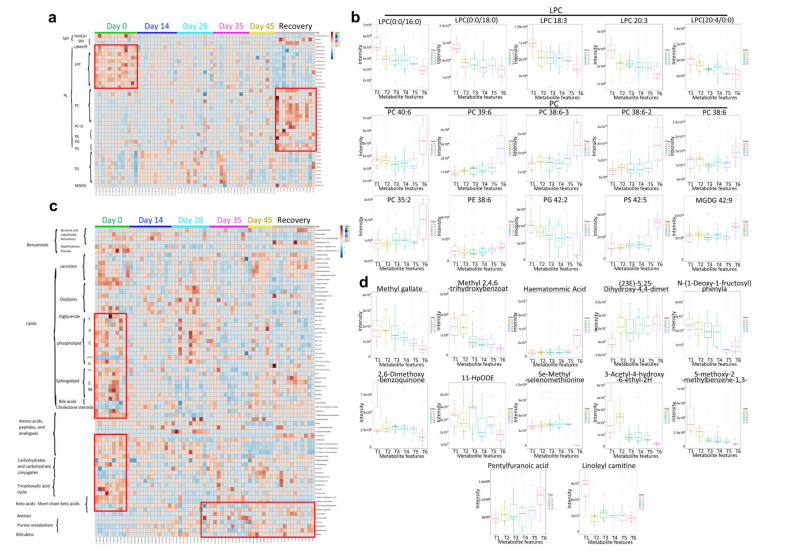
Human plasma lipidomics and metabolomics of vaccination-related or COVID-19-recovered samples and correlated analysis. (**a**,**b**) Heatmap. (**c**,**d**) Boxplot. T1 = Day 0, T2 = Day 14, T3 = Day 28, T4 = Day 35, T5 = Day 45, and T6 = Recovery.

## Data Availability

The data presented in this study are available in the article and Appendix A.

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
