# Peer review of "An Integrative Analysis of the Immune Features of Inactivated SARS-CoV-2 Vaccine (CoronaVac)"

_vaccines, 2022, doi:10.3390/vaccines10060878_

Round 1

Reviewer 1 Report

In this article the authors wanted to present a detailed in vitro immune cellular response and large-scale multi-omics analysis for peripheral blood mononuclear cells (PBMCs) from participants vaccinated with CoronaVac (Sinovac Life Sciences, Beijing, China) and recovered participants from COVID-19.

Several SARS-CoV-2 vaccines have been approved by a number of countries, including inactivated vaccines (such as CoronaVac). The authors point out that the use of an inactivated vaccine maintains the immunogenicity of the virus to induce protective immune responses. The CoronaVac vaccine has demonstrated good safety, immunogenicity, and efficacy in its clinical trials and does appear to be effective against the Delta variant infection. The authors claim that following 2 doses of the CoronaVac vaccine, an estimated efficacy exceeding the minimal threshold of 50% set by the World Health Organisation is achieved. The vaccine has been approved in 46 countries and 26 clinical trials for CoronaVac have been performed in eight countries demonstrating the utility of this vaccine.

Although the neutralising antibody titers decreased significantly in recipients within a few months of receiving the second dose of the vaccine, Xie et al. demonstrated that CoronaVac elicited a humoral immune response, including neutralising antibodies, to SARS- CoV-2 variants, and suggested that a third-dose boost may be beneficial for combating SARS-CoV-2 and its variants. The authors highlight that the in vitro cellular immune features of CoronaVac are still unclear and require more thorough studies to demonstrate the important role the third-dose boost plays in further inducing effective immune memory.

The authors note that recent large-scale multi-omics analyses including single-cell RNA sequencing, proteomics, metabolomics, and lipidomics, have been used to illustrate the immune features between COVID-19 patients and healthy individuals. They state that these studies demonstrated that integrating multi-omics with an in vitro immune response analysis could be applied to evaluate the protective effects of a vaccine, thereby accelerating vaccine development. Obviously, there is a real need for this type of research and scientific endeavour and so Jiang et al stated that the aim of their study was to integrate in vitro immune assays and large-scale multi-omics analyses in order to reveal the immune characteristics of CoronaVac. They investigated the single-cell landscape of PBMC samples from healthy participants at a number of timepoints before and after CoronaVac vaccinations and from control participants that had recovered from COVID-19 for approximately eight months. Proteomic, metabolomic, and lipidomic analyses were also performed for the serum or plasma samples in order to compare to the single-cell transcriptomics data that they had obtained.

The authors conclude that their results demonstrate that they have provided a comprehensive understanding of CoronaVac-specific in vitro immune features.

Main points and comments:

  1. Do the authors have any information regarding the Covid-19 participants in the study (ie what was the severity of their Covid infection and what was the duration of their original clinical signs; were they all age matched; what gender were they)? These pieces of information would help to shape this manuscript with some more detail.
  2. My main concern with this manuscript is the low number of participants in the various cohorts and the low number of samples tested by scRNASeq. Can the authors please comment on these points and suggest how these issues might be addressed?
  3. Line 82. Do the authors really mean an “anal swab sample” or is this a typing error?
  4. What is the rationale for selecting 21 to 54 year old participants for the vaccination study group?
  5. Why were the recovery group made up of individuals from different age groups (28 to 75 years) in comparison with the vaccination group?
  6. The group sizes used in this study were very small (13 participants and 12 participants). Can the authors please comment and clarify this point?
  7. Can the authors please clarify at what stage they were testing less than 25 participants as line 92 mentions that 5 vaccination participants withdrew from the study. This means a drop of 38% of participants from the vaccination group and a drop of 20% overall from the 25 original participants.
  8. Line 95 says “…….20-50 years” but on lines 79 and 83-84 there are different ages mentioned. Can the authors please make their statements consistent throughout the manuscript or clarify why there are different ages mentioned in different places?
  9. Line 199. Please can the authors explain what they mean by this sentence? Why “most” of the samples? What happened to the rest? If the authors are referring to the 20 participants that they ended up with after 5 members of the vaccination group withdrew, then that would mean 20% of the sample group were missing and so 80% would be left (not greater than 80% as stated on line 199). Please clarify this issue. Also refer to lines 158 to 160 looking at a similar point.
  10. Figure 1 is very misleading and needs to be adjusted to show where the cohort numbers changed during the protocol. Line 93 states that some blood samples were not collected during some of the timepoints but it is not clear where this happened and which timepoints were affected.
  11. Figure 1 and the legend need to be changed to reflect the fact that 5 people withdrew.
  12. How were the eleven representative participants chosen (line 217)?
  13. Please make Day 0 consistent throughout (not Day 00 as in some places such as Figure 1a and 1b, as well as lines 176, 299 and 329 as examples).
  14. Figure 2 legend states that vaccination was carried out at the “indicated timepoints” but there are no timepoints shown on the Figure to demonstrate when the vaccinations were done. Please clarify and adjust as required.
  15. Can the authors comment on the large number of naïve B cells shown in Figure 2b?
  16. Would it have been useful to collect PBMCs after day 45 to give a much later timepoint for analyses (Figure 2 and Figure 3)? Comparison of 8-month recovery convalescent samples and samples collected from vaccinated groups 45 days post vaccination 1- and 17-days post vaccination 2 are not comparable and are not going to give robust results. Can the authors comment please?
  17. Data availability. Line 190. Please state which repositories have been used.
  18. Figure 4 is particularly interesting with regards to positive gene expression responses in the PBMCs.
  19. Figure S3. Can the authors please comment on the 33 differentially expressed genes in the vaccinated group and the 62 differentially expressed genes in the recovery group (including the 23 differentially expressed genes that were found in both groups)?
  20. How were the participants picked to take part in Figure 4a? What were the entry criteria?
  21. Minor point. Some of the Figures are so small they are difficult to read such as Figures 5e, 6a, 7b and 7d.
  22. Can the authors explain why the latest timepoint was 45 days for the vaccinated participants? This does not seem to be a long enough period of time to allow for analysis of immune responses and the detection of neutralising antibodies in the samples that have been collected. Later timepoints (6 or 8 months), would allow for more accurate assessment of the responses gained following vaccination. Can the authors please comment on these points?
  23. Do the authors have any indication as to whether the CoronaVac vaccine is effective against multiple variants of SARS-CoV-2?
  24. The authors state that they have been able to supply a “…comprehensive understanding of CoronaVac-specific in vitro immune features” but I query that based on the evidence they provide from using a boosting immunisation at 28 days. There needs to be a much larger time frame for the assessment of immune features….much longer than the 45 day timepoint for the vaccinated group. The recovery controls were assessed 8 months after infection and so the authors are not comparing like with like. Can they please comment on this?
  25. This study is a very worthy cause and I think the authors may want to look at some recent publications in a similar area of research to assist their progress.

Suggested reading:

Glück V, Tydykov L, Mader AL, et al. Humoral immunity in dually vaccinated SARS-CoV-2-naïve individuals and in booster-vaccinated COVID-19-convalescent subjects [published online ahead of print, 2022 Apr 11]. Infection. 2022;1-7. doi:10.1007/s15010-022-01817-8

Wagner R, Meißner J, Grabski E, Sun Y, Vieths S, Hildt E. Regulatory concepts to guide and promote the accelerated but safe clinical development and licensure of COVID-19 vaccines in Europe. Allergy. 2022;77(1):72-82. doi:10.1111/all.14868.

Author Response

Review1:

In this manuscript, Jiang and colleagues have shown the immunological characteristics and efficacy of the inactivated SARS-CoV-2 corona vaccine (SinoVac), using the extensive in-vitro and RNA-sequence analysis with PBMCs acquired from the volunteers immunized or present in being recovered. This study possesses a value addressing the scientific indication for better understanding the efficacy of the inactivated corona vaccine on eliciting the antigen-specific humoral and T-cell immunity of the inactivated corona vaccine, although the local immunity induced at the regions of upper respiratory tissues remains still equivocal. This manuscript is well written and conveys authors’ scientific message well via integrated approaches. Here are my additional comments.

Response: We thank the reviewer’s positive comments.

  1. Figure 2 legend needs to provide more information on statistical explanation about significant alterations (c-e) although some descriptions are shown in the Methods. For example, the asterisks and controls used in comparisons should be defined. Also, the authors need to clarify if statistical significance shown in the graphs of c, d, and e is correctly assessed.
    Response: We added the related descriptions as the reviewer’s suggestion. We clarified the statistical significance.
  2. In the Figure 2c, dot plots for Tfh cells show Th1-like, Th2-like and Th17-like cells with regards to differential expression levels of chemokine receptors CCR6 and CXCR3. Authors need to provide any scientific evidence on this classification with any evidential description with relevant references, if it hasn’t been shown in the text.In the flow-cytometry dot plots of Figure 2b-d, the labels of X- and Y-axis are too small. They need to be enlarged.
    Response: We rewrite this part as suggested and enlarged the labels of X- and Y-axis.
  3. Regarding the description on lines 276-278 for data explaining Figure 3, it is written that current data support outcomes of clinical studies with the corona vaccines in different countries. The authors need to provide any evidential clue or references for these statements if available.
    Response: We modified the statement and cited the related references.
  4. In Figure 6a, the labels of reactome pathways in the center of each hierarchy also need to be enlarged. The red-colored circles should be defined in the legends as well.
    Response: We enlarged it and defined it.
  1. In lines 358-359, the authors need to specify “different immune cells” in the sentence.
    Response: We specified them. The immune cells here include B cells, T cells (total, CD4+ or CD8+), NK cells and myeloid cells. We have revised it accordingly.

Reviewer 2 Report

In this manuscript, Jiang and colleagues have shown the immunological characteristics and efficacy of the inactivated SARS-CoV-2 corona vaccine (SinoVac), using the extensive in-vitro and RNA-sequence analysis with PBMCs acquired from the volunteers immunized or present in being recovered. This study possesses a value addressing the scientific indication for better understanding the efficacy of the inactivated corona vaccine on eliciting the antigen-specific humoral and T-cell immunity of the inactivated corona vaccine, although the local immunity induced at the regions of upper respiratory tissues remains still equivocal. This manuscript is well written and conveys authors’ scientific message well via integrated approaches. Here are my additional comments.

  • Figure 2 legend needs to provide more information on statistical explanation about significant alterations (c-e) although some descriptions are shown in the Methods. For example, the asterisks and controls used in comparisons should be defined. Also, the authors need to clarify if statistical significance shown in the graphs of c, d, and e is correctly assessed.
  • In the Figure 2c, dot plots for Tfh cells show Th1-like, Th2-like and Th17-like cells with regards to differential expression levels of chemokine receptors CCR6 and CXCR3. Authors need to provide any scientific evidence on this classification with any evidential description with relevant references, if it hasn’t been shown in the text.
  • In the flow-cytometry dot plots of Figure 2b-d, the labels of X- and Y-axis are too small. They need to be enlarged.
  • Regarding the description on lines 276-278 for data explaining Figure 3, it is written that current data support outcomes of clinical studies with the corona vaccines in different countries. The authors need to provide any evidential clue or references for these statements if available.
  • In Figure 6a, the labels of reactome pathways in the center of each hierarchy also need to be enlarged. The red-colored circles should be defined in the legends as well.
  • In lines 358-359, the authors need to specify “different immune cells” in the sentence.

Author Response

Review2:

In this article the authors wanted to present a detailed in vitro immune cellular response and large-scale multi-omics analysis for peripheral blood mononuclear cells (PBMCs) from participants vaccinated with CoronaVac (Sinovac Life Sciences, Beijing, China) and recovered participants from COVID-19.

Several SARS-CoV-2 vaccines have been approved by a number of countries, including inactivated vaccines (such as CoronaVac). The authors point out that the use of an inactivated vaccine maintains the immunogenicity of the virus to induce protective immune responses. The CoronaVac vaccine has demonstrated good safety, immunogenicity, and efficacy in its clinical trials and does appear to be effective against the Delta variant infection. The authors claim that following 2 doses of the CoronaVac vaccine, an estimated efficacy exceeding the minimal threshold of 50% set by the World Health Organisation is achieved. The vaccine has been approved in 46 countries and 26 clinical trials for CoronaVac have been performed in eight countries demonstrating the utility of this vaccine.

Although the neutralising antibody titers decreased significantly in recipients within a few months of receiving the second dose of the vaccine, Xie et al. demonstrated that CoronaVac elicited a humoral immune response, including neutralising antibodies, to SARS- CoV-2 variants, and suggested that a third-dose boost may be beneficial for combating SARS-CoV-2 and its variants. The authors highlight that the in vitro cellular immune features of CoronaVac are still unclear and require more thorough studies to demonstrate the important role the third-dose boost plays in further inducing effective immune memory.

The authors note that recent large-scale multi-omics analyses including single-cell RNA sequencing, proteomics, metabolomics, and lipidomics, have been used to illustrate the immune features between COVID-19 patients and healthy individuals. They state that these studies demonstrated that integrating multi-omics with an in vitro immune response analysis could be applied to evaluate the protective effects of a vaccine, thereby accelerating vaccine development. Obviously, there is a real need for this type of research and scientific endeavour and so Jiang et al stated that the aim of their study was to integrate in vitro immune assays and large-scale multi-omics analyses in order to reveal the immune characteristics of CoronaVac. They investigated the single-cell landscape of PBMC samples from healthy participants at a number of timepoints before and after CoronaVac vaccinations and from control participants that had recovered from COVID-19 for approximately eight months. Proteomic, metabolomic, and lipidomic analyses were also performed for the serum or plasma samples in order to compare to the single-cell transcriptomics data that they had obtained.

The authors conclude that their results demonstrate that they have provided a comprehensive understanding of CoronaVac-specific in vitro immune features.

Response: We are grateful to the reviewer’s positive comments.

Main points and comments:

  1. Do the authors have any information regarding the Covid-19 participants in the study (ie what was the severity of their Covid infection and what was the duration of their original clinical signs; were they all age matched; what gender were they)? These pieces of information would help to shape this manuscript with some more detail.
    Response: We have updated the information of participants in Figure 1, which includes the gender, age, withdrawing time points and recovered days of participants.
  2. My main concern with this manuscript is the low number of participants in the various cohorts and the low number of samples tested by scRNASeq. Can the authors please comment on these points and suggest how these issues might be addressed?
    Response: The sample size was vastly limited by the high cost of immunoassays and multi-omics, especially scRNA-seq. Despite the small sample size, we still observed some significant differences as shown in Figures 1, 2 and 3, which may be statistically adequate. Besides, scRNA-seq was a sensitive method for conveying multi-dimensional gene expression profiles and could reveal the subtle transcriptomic changes before and after vaccination according to the t-SNE clustering (Figure S1, good clustering within groups, distinct clustering between groups). We revised and addressed these points in lines 289-290, 308-310 and 323-326.
  3. Line 82. Do the authors really mean an “anal swab sample” or is this a typing error?
    Response: Thank you for pointing out our typing error. We removed it.
  4. What is the rationale for selecting 21 to 54 year old participants for the vaccination study group?
    Response: We planned to enroll adult or elder participants in the vaccination group. The 21-54 years old was not intended, but the real age range of the participants in this group. We removed this ambiguity in line 81.
  5. Why were the recovery group made up of individuals from different age groups (28 to 75 years) in comparison with the vaccination group?
    Response: The 28-75 years old was the real age range of the participants in the recovery group (see the updated information of participants in Figure 1). The participants of the vaccination group were slightly younger (real age range: 21-54 years old), because younger male participants are generally more willing as volunteers and more fit for blood sampling and vaccination. To make a good logic, we unified the rationale to selecting adult or elder participants for both groups as revised in lines 81 and 85.
  6. The group sizes used in this study were very small (13 participants and 12 participants). Can the authors please comment and clarify this point?
    Response: On one hand, this study was completed in the early period of the COVID-19 before the CoronaVac had been approved and the booster needle program, and some persons already withdrew this project. On the other hand, we considered the cost of this project, which discouraged larger sample sizes. Finally, the immunoassays demonstrated significant differences with samples from these 13+12 participants (Figures 1, 2 and 3). We have explained it in the corresponding position of the revised manuscript (lines 289-290).
  7. Can the authors please clarify at what stage they were testing less than 25 participants as line 92 mentions that 5 vaccination participants withdrew from the study. This means a drop of 38% of participants from the vaccination group and a drop of 20% overall from the 25 original participants.
    Response: Five vaccination participants got bored with the regular blood sampling and withdrew from the study. The withdrawing stage was listed in the updated information of participants in Figure 1. A total of 55 samples were collected from 13 participants in the vaccination group, including the early samples of the 5 withdrawing participants.
  8. Line 95 says “…….20-50 years” but on lines 79 and 83-84 there are different ages mentioned. Can the authors please make their statements consistent throughout the manuscript or clarify why there are different ages mentioned in different places?
    Response: We unified the statements as “adult or elder participants” and avoided unnecessary age ranges. The previous written age ranges just specified the actual distribution of ages in a group, not the rationale of enrolling participants.
  9. Line 199. Please can the authors explain what they mean by this sentence? Why “most” of the samples? What happened to the rest? If the authors are referring to the 20 participants that they ended up with after 5 members of the vaccination group withdrew, then that would mean 20% of the sample group were missing and so 80% would be left (not greater than 80% as stated on line 199). Please clarify this issue. Also refer to lines 158 to 160 looking at a similar point.
    Response: A total of 67 blood samples were collected in this study (see the updated part of Figure 1), including 55 samples from 13 participants in the vaccination group and 12 samples from 12 participants in the recovery group. However, some samples were used up after performing some experiments. The analyzed sample number in each experiment were shown below: serum antibody assay (67/67), lipidomics (67/67), metabolomics (64/67), proteomics (59/67), scRNA-seq (11/67). We clarified this in the final manuscript.
  10. Figure 1 is very misleading and needs to be adjusted to show where the cohort numbers changed during the protocol. Line 93 states that some blood samples were not collected during some of the timepoints but it is not clear where this happened and which timepoints were affected.
    Response: Five participants withdrew at varied time points in the vaccination group for poor compliance or getting bored with blooding sampling. We have updated the withdrawing stage in the participant information in Figure 1, and modified the descriptions in line 94-95 accordingly.
  11. Figure 1 and the legend need to be changed to reflect the fact that 5 people withdrew.
    Response: We have updated Figure 1 and its legend.
  12. How were the eleven representative participants chosen (line 217)?
    Response: After balancing the cost of scRNA-seq and the sample size, we determined to perform scRNA-seq on two vaccination participants (H1 and H2, each with 4 time points) and three recovery participants (R1, R2 and R3), which made up 11 scRNA-seq samples. Here, H1 and H2 were randomly selected from the 13 vaccination participants before performing the assays. R1, R2 and R3 were randomly selected from the 12 recovered patients. We changed the descriptions accordingly in lines 222-225.
  13. Please make Day 0 consistent throughout (not Day 00 as in some places such as Figure 1a and 1b, as well as lines 176, 299 and 329 as examples).
    Response: We unified this annotation to Day 0.
  14. Figure 2 legend states that vaccination was carried out at the “indicated timepoints” but there are no timepoints shown on the Figure to demonstrate when the vaccinations were done. Please clarify and adjust as required.
    Response: We corrected it.
  15. Can the authors comment on the large number of naïve B cells shown in Figure 2b?
    Response: We check the original data and find the percentage of CD27-IgD+ naïve B cells vary from about 40% to 70%. This result coincides with other publications (Clin. Exp. Immunol. 162, 271–279 (2010); Arthritis Res. Ther. 21, 1–11 (2019); Front. Cell. Infect. Microbiol. 2, 128 (2012).), and we added the related description and cited these references.
    For example, in figure 2 of the first publication (Clin. Exp. Immunol. 162, 271–279 (2010)), they calculated the relative frequencies of different B cell subsets in a cohort of 221 individuals ranging from neonates to adults by flow cytometry. The naïve B population (CD27-IgD+) decline during age and vary significantly among each period. 40-75% seems to be a range that most individual fall in during age 25-50. In figure 3A of the second publication (Arthritis Res. Ther. 21, 1–11 (2019)), in the baseline group, the bar indicates that the value varied from about 40-80%.
    In figure 1B of the third publication (Front. Cell. Infect. Microbiol. 2, 128 (2012)), the value varied from about 60-85%. In our research, we performed paired-sample T test to reduce error origin from individual difference.
  16. Would it have been useful to collect PBMCs after day 45 to give a much later timepoint for analyses (Figure 2 and Figure 3)? Comparison of 8-month recovery convalescent samples and samples collected from vaccinated groups 45 days post vaccination 1- and 17-days post vaccination 2 are not comparable and are not going to give robust results. Can the authors comment please?
    Response: In this study, we aim to analyze the early immune characteristics and gene expression profiles induced the CoronaVac to complement to its clinical data, and this study was completed in the early period of the COVID-19 before the CoronaVac had been approved and the booster needle program, and some persons already withdrew this project. On the other hand, we considered the cost of this project. We have explained it in the corresponding position of the revised manuscript (lines 528-532).
  17. Data availability. Line 190. Please state which repositories have been used.   Response: We have uploaded the data to the figshare repository (DOI: 10.6084/m9.figshare.19808818).
  18. Figure 4 is particularly interesting with regards to positive gene expression responses in the PBMCs.
    Response: Thank you. The scRNA-seq is a powerful technique in monitoring the transcriptomic changes for each immune cell type, and we used it to explore the response of vaccinations, which is a feature of this manuscript.
  19. Figure S3. Can the authors please comment on the 33 differentially expressed genes in the vaccinated group and the 62 differentially expressed genes in the recovery group (including the 23 differentially expressed genes that were found in both groups)?
    Response: These differentially expressed genes were considered to be highly associated to the CoronaVac vaccination or SARS-CoV-2 infection. We have added some comments for these differentially expressed genes in lines 365-368, 371-372 and 378-379.
  20. How were the participants picked to take part in Figure 4a? What were the entry criteria?
    Response: Considering the high cost of scRNA-seq, we decided to cut down the sample size to 11 PBMC samples. Further, we determined the number of participants and samples in the vaccination group (2 participants, each with 4 time points) and the recovery group (3 participants, each with 1 time point). Finally, we randomly selected 2 adult participants from the vaccination cohort and 3 adult participants from the recovery cohort (lines 97-98, Figure 4 legend).
  21. Minor point. Some of the Figures are so small they are difficult to read such as Figures 5e, 6a, 7b and 7d.
    Response: We enlarged them.
  22. Can the authors explain why the latest timepoint was 45 days for the vaccinated participants? This does not seem to be a long enough period of time to allow for analysis of immune responses and the detection of neutralising antibodies in the samples that have been collected. Later timepoints (6 or 8 months), would allow for more accurate assessment of the responses gained following vaccination. Can the authors please comment on these points?
    Response: In this study, we aim to analyze the early immune characteristics and gene expression profiles induced by the CoronaVac to complement to its clinical data, and this study was completed in the early period of the COVID-19 before the CoronaVac had been approved and the booster needle program, and some persons already withdrew this project. On the other hand, we considered the cost of this project. We have explained it in the corresponding position of the revised manuscript (lines 528-532).
  23. Do the authors have any indication as to whether the CoronaVac vaccine is effective against multiple variants of SARS-CoV-2?
    Response: The two-dose vaccination effective against the Delta variant, such as B.1.351, also known as 501Y.V2 infection has been reported (Emerg. Microbes Infect. 2021, 1–32; Cell Res. 2021, 31, 732–741). Although, CoronaVac can not be efficiently against beta and Omicron variant of SARS-CoV-2 (https://www.rojakpot.com/pfizer-vs-sinovac-omicron-variant/), the vaccine against Omicron variant by Sinovac Life Sciences has been developed and approved. We added the related description in the Discussion section.
  24. The authors state that they have been able to supply a “…comprehensive understanding of CoronaVac-specific in vitro immune features” but I query that based on the evidence they provide from using a boosting immunisation at 28 days. There needs to be a much larger time frame for the assessment of immune features….much longer than the 45 day timepoint for the vaccinated group. The recovery controls were assessed 8 months after infection and so the authors are not comparing like with like. Can they please comment on this?
    Response: In this study, we aim to analyze the early immune characteristics and gene expression profiles induced by the CoronaVac to complement to its clinical data, and this study was completed in the early period of the COVID-19 before the CoronaVac had been approved and the booster needle program, and some persons already withdrew this project. On the other hand, we considered the cost of this project. We have explained it in the corresponding position of the revised manuscript (lines 528-532).
  25. This study is a very worthy cause and I think the authors may want to look at some recent publications in a similar area of research to assist their progress.
    Response: Thank you very much. We have read and cited the publications in the discussion and introduction, respectively.

Round 2

Reviewer 1 Report

The revised manuscript is a vast improvement on the original and is much clearer now for the reader. The authors have clearly addressed the questions and attempted to justify the manuscript. They have improved many of the Figure legends and have added additional text to enhance the clarity of the manuscript.

There are still a few minor issues (some of which I have listed below):

 1. “….blooding sampling….. “ on line 93 needs the English correcting.

2. There are a couple of words missing in line 94 ……“…for the rest of the timepoints.”

3. I am pleased to see that section 2.10 has been clarified fully.

4. Figure 1 legend still has some English language issues such as:

“(a) Diagram showing the number and information of participants in the study”

This should read (a) Diagram showing the number of, and information of regarding, participants in the study.

“The participant information was shown on the right,……”

This should read “The participant information is shown on the right,…..”

5. Section 3.2 has been improved.

6. Figure 2 legend has been improved and clarified and the details for the stats have been added.

7. The additional sentences have clarified the text throughout the manuscript and have improved the details that have been given. Thank you.

8. Figure 6 legend still needs a little bit of editing to correct the English.

9. Figure 7 has been made much clearer. Thank you.

10. Line 482 needs a 0 taking off the end of “Day 00” as previously mentioned for other places in the text.

11. Lines 522 to 530 have attempted to explain the reasons behind the study and the limitations that exist, and I fully appreciate the issues involved with a study of this nature.

I note the addition of some extra references and the manuscript has been improved with the changes that have been made and the issues that have been addressed.

Author Response

 Point-to-Point rebuttal to reviewer 1’s comments

We are grateful to the editor and reviewer for the positive comments again. We have addressed all the concerns. The followings are our Point-to-Point answers for the questions. Questions and answers are in black and blue, respectively. The revised texts are marked in red in the manuscript.

Review1:

The revised manuscript is a vast improvement on the original and is much clearer now for the reader. The authors have clearly addressed the questions and attempted to justify the manuscript. They have improved many of the Figure legends and have added additional text to enhance the clarity of the manuscript.

Response: We are grateful to the reviewer’s comments.

There are still a few minor issues (some of which I have listed below):

  1. “….blooding sampling….. “ on line 93 needs the English correcting.

Response: We corrected it.

  1. There are a couple of words missing in line 94 ……“…for the rest of the timepoints.”

Response: We added the missing words.

  1. I am pleased to see that section 2.10 has been clarified fully.

Response: Thank you very much.

  1. Figure 1 legend still has some English language issues such as:

“(a) Diagram showing the number and information of participants in the study”

This should read (a) Diagram showing the number of, and information of regarding, participants in the study.

“The participant information was shown on the right,……”

This should read “The participant information is shown on the right,…..”

Response: We revised it as the reviewer suggestion.

  1. Section 3.2 has been improved.

Response: Thank you very much.

  1. Figure 2 legend has been improved and clarified and the details for the stats have been added.

Response: Thank you very much.

  1. The additional sentences have clarified the text throughout the manuscript and have improved the details that have been given. Thank you.

Response: Thank you very much.

  1. Figure 6 legend still needs a little bit of editing to correct the English.

Response: We edited it again.

  1. Figure 7 has been made much clearer. Thank you.

Response: Thank you very much.

  1. Line 482 needs a 0 taking off the end of “Day 00” as previously mentioned for other places in the text.

Response: We corrected it.

  1. Lines 522 to 530 have attempted to explain the reasons behind the study and the limitations that exist, and I fully appreciate the issues involved with a study of this nature.

Response: Thank you very much.

I note the addition of some extra references and the manuscript has been improved with the changes that have been made and the issues that have been addressed.

Response: Thank you very much.
